# Argon Adsorption on Cationic Gold Clusters Au*_n_*^+^ (*n* ≤ 20)

**DOI:** 10.3390/molecules26134082

**Published:** 2021-07-04

**Authors:** Piero Ferrari, Ewald Janssens

**Affiliations:** Quantum Solid-State Physics, Department of Physics and Astronomy, KU Leuven, 3001 Leuven, Belgium; piero.ferrari@kuleuven.be

**Keywords:** gold clusters, argon, adsorption energies, mass spectrometry, density functional theory

## Abstract

The interaction of Au*_n_*^+^ (*n* ≤ 20) clusters with Ar is investigated by combining mass spectrometric experiments and density functional theory calculations. We show that the inert Ar atom forms relatively strong bonds with Au*_n_*^+^. The strength of the bond strongly varies with the cluster size and is governed by a fine interplay between geometry and electronic structure. The chemical bond between Au*_n_*^+^ and Ar involves electron transfer from Ar to Au, and a stronger interaction is found when the Au adsorption site has a higher positive partial charge, which depends on the cluster geometry. Au_15_^+^ is a peculiar cluster size, which stands out for its much stronger interaction with Ar than its neighbors, signaled by a higher abundance in mass spectra and a larger Ar adsorption energy. This is shown to be a consequence of a low-coordinated Au adsorption site in Au_15_^+^, which possesses a large positive partial charge.

## 1. Introduction

As a bulk material, gold is the noblest of all metals [1]. At the nanoscale, however, gold becomes reactive towards several molecules, and can selectively catalyze certain reactions, which has attracted broad scientific attention [2,3,4,5]. Intriguingly, gold cationic clusters are also reactive towards noble gas atoms [6]. The inertness of noble gases was challenged by the work of P. Pyykkö, who in 1995 proposed that the bond between Au^+^ ions with different noble gases (Ng = Ar, Kr, Xe) has a partially covalent character, and cannot merely be ascribed to the expected weak dispersion forces [7]. While this claim was controversial at that time [8,9,10], it is nowadays accepted, and noble gas chemistry has become an active research field, with Au-based systems drawing significant attention [6,11].

The properties of gold clusters differ significantly from those of copper and silver coinage metal clusters [12,13,14,15]. The intriguing properties of gold clusters are not restricted to their reactivity, but also to unique ground-state geometries [16], optical absorption [17], relative stability [18] and radiative cooling [19], and are largely related to the relativistic nature of the Au atom [20]. One consequence of the relativistic effects in Au is an enhanced electron affinity [20], if compared, for instance, with Ag and Cu. The high electron affinity was shown to favor electron transfer from the Ng atom to Au in the AuNg^+^ dimer and therefore the formation of a relatively strong bond [21]. 

Using the Au_3_^+^ trimer as a model system, the interaction between Au and Ar was investigated by Fielicke and collaborators, using a combination of infrared spectroscopy and density functional theory (DFT) calculations [22]. Ar was shown to be heavily involved in the highest frequency vibrational modes of Au_3_Ar*_m_*^+^ complexes, something that was attributed to a relativistically enhanced covalency of the Au-Ar interaction. This conclusion was supported by performing experiments on Ag_3−*n*_Au*_n_*^+^ clusters, where the Ar bonding decreased with increasing silver concentration. Subsequent work by the same authors demonstrated similar characteristics of the interaction of Ar with Au_4_^+^ and Au_5_^+^, namely, electron transfer from Ar to the cationic gold clusters and the formation of a strong bond of partial covalent character [23].

The attachment of Ar (and other noble gas elements) to larger cationic gold clusters has been reported. For example, Ar has been used as a tagging/spectator atom when investigating the geometries and the optical absorption spectra of pure and doped gold clusters [24,25,26,27,28]. Moreover, in investigations of the optical absorption of Au*_n_*^+^ clusters in the *n* = 13–20 size range, Xe was employed as the tagging element instead of Ar, as the interaction of the latter noble gas element was seen to decrease with cluster size and therefore, it was easier to form the complexes with Xe [29]. Nevertheless, the nature of the interaction between the noble gas atoms with Au*_n_*^+^, in a broad size range, has not been explained. For instance, Au_15_^+^ has been noticed to adsorb a larger amount of Ar than the closest size neighbors [28]. This feature is unexpected, as Au_15_^+^ possesses a closed electronic configuration, and therefore, might be expected to be less reactive than the open-shell neighboring sizes.

In this work, we address the interaction between Au*_n_*^+^ clusters with Ar, by performing density functional theory calculations in the extended *n* ≤ 20 size range, in combination with mass spectrometric experiments. We demonstrate the formation of a (partial) chemical bond between Ar and the clusters, which is a consequence of an electron charge donation from Ar to Au. The strength of the cluster−argon interaction is strongly size dependent, and is well correlated with the geometry of the Au*_n_*^+^ clusters. 

## 2. Methods

### 2.1. Experimental

The gold clusters were produced with the laser ablation technique [30], in a setup that is described in detail in Reference [31]. In short, a bulk gold plate was ablated by a tightly focused nanosecond Nd:YAG laser (second harmonic, 532 nm), and the generated plasma was cooled by a pulse of ultrapure He gas, added into the cluster source at a pressure of 8 bar. The mixture gold-He expands supersonically into vacuum, generating a molecular beam that was collimated by a 1 mm skimmer and was analyzed by reflectron time-of-flight mass spectrometry (Δ*m*/*m* ≈ 1000). The laser power density and the time delay between the gas injection and the ablation were optimized to produce Au*_n_*^+^ clusters in the *n* ≤ 20 size range. Neutral and anionic species were also formed by this technique, however, they were not detectable by the mass spectrometer in the current settings. In order to make argon complexes, the He gas was mixed with 2% of Ar and the source was cooled to 250 K, resulting in Au*_n_*Ar*_m_*^+^ complexes with up to *m* = 5, depending on cluster size. If a lower source temperature is used, the clusters are covered with even more Ar, whereas the amount of Ar attachment was minor for the larger clusters at higher source temperatures.

### 2.2. Computational

Density functional theory calculations were performed to complement the mass spectrometric findings, covering the entire range of Au*_n_*Ar*_m_*^+^ clusters with *n* = 3–20 and *m* ≤ 5. All calculations were conducted employing the ORCA 4.2.1 software package [32], by using the PBE exchange-correlation functional [33], in conjunction with the Def2-TZVPP basis set [34]. The Stuttgart Def2-ECP pseudopotential was employed for Au, which treats explicitly 19 valence electrons [34]. Scalar relativistic effects are included in the ECP. All the electrons of Ar were considered in the calculations. Dispersion forces were accounted for via the D3BJ dispersion-correction [35]. This level of theory was shown to be accurate and computationally efficient for describing the interactions in Au^+^-Ar*_m_* (*m* ≤ 4) [21]. The geometries of the bare Au*_n_*^+^ (*n* = 3–20) clusters were adopted from previous works. Infrared multiple photon dissociation spectroscopy was used in conjunction with DFT calculations to determine the geometries of the *n* = 3–9 sizes in Reference [24], whereas an extensive global search of low-energy isomers was performed for the bare clusters up to *n* = 20 in Reference [36]. Here, we have adopted the bare cluster geometries determined in Reference [24] up to *n* = 9, and those from Reference [36] in the *n* = 10–20 size range. We note that in Reference [24], the same level of theory as applied here was selected and a benchmark analysis showed that it accurately describes the vibrational modes of Au*_n_*Ar*_m_*^+^ clusters. To locate the preferred Ar adsorption sites, Ar was then placed at all possible atop coordination positions, followed by geometry optimization. Starting geometries with other Ar coordination sites were tested as well. Nevertheless, atop adsorption sites always resulted in lower total energies. 

The adsorption energy (*E_ads_*) of the *m*th attached Ar atom in the Au*_n_*Ar*_m_*^+^ complex was calculated by Equation (1):(1)$$E_{ads}\left(n,m\right)=E\left({\mathrm{Au}}_{n}{\mathrm{Ar}}_{m}{}^{+}\right)-E\left({\mathrm{Au}}_{n}{\mathrm{Ar}}_{m-1}{}^{+}\right)-E\left(\mathrm{Ar}\right)$$
where *E* is the (absolute) total energy of the cluster in parenthesis. The reported adsorption energies have not been corrected for zero point energy. Moreover, Wiberg bond indexes and electron localization functions (ELF) have been calculated using the Multiwfn software package [37].

## 3. Results

The mass spectrometric results are summarized in Figure 1, where the fractional distribution of Ar in the Au*_n_*Ar*_m_*^+^ cluster is presented. These fractions are defined by Equation (2), with *I* the area of the peak corresponding to the complex within parenthesis in the mass spectrum.
(2)$$F_{n}\left(m\right)=\frac{I\left({\mathrm{Au}}_{n}{\mathrm{Ar}}_{m}{}^{+}\right)}{\sum_{i}I\left({\mathrm{Au}}_{n}{\mathrm{Ar}}_{i}{}^{+}\right)}$$

There are several interesting observations from Figure 1. (i) First of all, more Ar adsorbs on the smaller cationic clusters. More specifically, up to *n* = 7, the intensity of bare Au*_n_*^+^ is very small, (i.e., the difference between 1 and the stacked bar chart of the argon fractions for *m* = 1–5 and *n* = 3–6, and for *m* = 1–6 and *n* = 7) as is the fraction of the complexes with a single Ar atom, *m* = 1. (ii) Second, for these smaller clusters (*n* ≤ 7), the *F_n_*(*m*) fractions with *m* = 2–5 are strongly size-dependent. Au_3_^+^ and Au_6_^+^ mainly adsorb three Ar atoms, while Au_4_^+^ and Au_5_^+^ primarily attach four. For Au_7_^+^, the *F*_7_(*m*) fractions with *m* = 2, 3, 4, 5 and 6 have about the same magnitude. (iii) Au_11_^+^ adsorbs more Ar atoms than its neighboring sizes and there is a gradual decrease in Ar fraction from *n* = 11 onwards. (iv) From *n* ≥ 14 onwards, only a single Ar atom is adsorbed on the clusters. (v) Last, the cluster *n* = 15 stands for its very high fraction of a single Ar atom.

In order to rationalize those mass spectrometric results, DFT calculations were performed on the Au*_n_*Ar*_m_*^+^ (*n* = 3–20; *m* ≤ 5) clusters. The geometries of the complexes that have a single Ar atom adsorbed are depicted in Figure 2. In all cases, Ar adopts an atop coordination in the lowest-energy configuration, and its presence hardly affects the geometries of the bare Au*_n_*^+^ clusters [24,36]. The possibility that isomers with a different adsorption site for Ar are present in the molecular beam, or that there is fluxionality between different adsorption sites, cannot be excluded [27]. As an example, isomers with a different Ar adsorption site for Au_4_Ar^+^ and Au_6_Ar^+^ are presented in the supporting information. The relative energies of these isomers are high enough to assume that they will not be present in large amounts in the molecular beam, however this cannot be excluded for other sizes. The results provided in the following support our analysis only considering the lowest energy configurations. 

For the single Ar lowest energy isomers up to *n* = 7 the Au framework is planar; for all larger sizes the metal frameworks are three-dimensional. Therefore, a first correspondence between the Au*_n_*^+^ cluster geometries and the experimental Ar fractions is already identified, namely the amount of adsorbed Ar decreases sharply at the transition size from planar to three-dimensional clusters, i.e., after Au_7_^+^. A similar effect was seen in anionic Au*_n_*^−^ species, where the size-dependent attachment of Ar was found to decrease abruptly once the clusters were three-dimensional [26]. The particular geometry adopted by the Au_15_^+^ cluster may also provide a hint for the higher *F*_15_(1) fraction. It is formed by a double layer of atoms, with a Au atom located on one layer [38]. As shown in Figure 2, this low-coordinated Au atom in Au_15_^+^ is the adsorption site of the first Ar. 

The observation that Ar adsorbs on atop coordination sites in the Au*_n_*^+^ clusters may help in understanding the size-dependent Ar fractions in the *n* ≤ 7 size range. Indeed the triangular shape of the Au_3_^+^ cluster with three equivalent atop adsorption sites is consistent with the large *F*_3_(3) fraction. Similarly, the rhombic geometry of Au_4_^+^ has four possible atop sites for Ar adsorption, and the higher fraction for this size is *F*_4_(4). Au_5_^+^ has a bowtie-like geometry with four equivalent low-coordinated sites, consistent with the observed high *F*_5_(4) fraction. The triangular shape of Au_6_^+^ has three corners on which atop Ar adsorption is possible, and *F*_6_(3) is the largest fraction (a second isomer of Au_6_^+^ has been seen in Reference [24] from which a similar conclusion can be drawn, as it also has three corners). Finally, Au_7_^+^ has a hexagonal shape with a Au atom at its center, hence six equivalent adsorption sites. In this cluster, similar fractions are observed for 2, 3, 4, 5 and 6 Ar atoms (it is the only cluster for which we find an *m* value as high as 6). Some examples of geometries for Au*_n_*^+^ clusters with multiple attached Ar atoms are presented in Figure 3 (all *m* = 0–5 structures are available as Appendix A). We remark that the discussion of Ar fractions in relation to the Au*_n_*^+^ cluster geometries is illustrative, but that further analysis of energetics as well as electronic structures are required to better understand the features observed experimentally.

Using Equation (1), the adsorption energies (*E_ads_*) of the *m*th adsorbed Ar atom in Au*_n_*Ar*_m_*^+^ were calculated. For this, the optimized geometries of the Au*_n_*Ar*_m_*^+^ clusters were employed, which are shown in Figure 2 for *m* = 1 (some complexes with larger *m* are presented in Figure 3 and the others are available as Appendix A). The obtained *E_ads_* energies are given in Figure 4. Several, if not all, experimental observations about the Ar fractions can be understood based on these adsorption energies. (a) The adsorption energies show an overall decrease with Au*_n_*^+^ size, in agreement with the Ar fractions being higher for the smaller clusters. (b) For *n* = 3, there is a sudden decrease in *E_ads_* at *m* = 4, while for *n* = 4 and 5, the values remain high up to *m* = 4. For *n* = 6, *E_ads_* is low for *m* = 4, while for *n* = 7, the energies are relatively high up to *m* = 6 (for *m* = 6, *E_ads_* is 0.16 eV). (c) There are local maxima in *E_ads_* for *m* = 1 at *n* = 11 and *n* = 15. (d) The adsorption energies slightly increase around *n* = 17, decreasing again at *n* = 19. Hence, the DFT calculations reproduce outstandingly well the mass spectrometric observations. 

## 4. Analysis

The Hirshfeld partial charge at the Au adsorption site in bare Au*_n_*^+^ and that of Ar in Au*_n_*Ar^+^ are presented in panel (a) of Figure 5. Qualitatively similar results were obtained with Löwdin partial charges analysis, but are not further discussed. The positive partial charge of the reactive Au site is +0.33 e for Au_3_^+^ and shows an overall decrease with size *n*, which is not surprising since with increasing size the total +1 charge of the cluster is distributed over more atoms. The decrease, however, is not homogeneous. In particular, the higher partial charge of the Ar adsorption site of Au_15_^+^ stands out as well as the pronounced minimum of Au_16_^+^. Clearly, there is a strong correlation between the charge of the gold adsorption site before Ar adsorption with that of Ar after adsorption: the adsorbed Ar is more positively charged if the adsorption site had a larger positive charge. The correlation between the Hirshfeld partial charges and *E_ads_* (and therefore with the experimental results) is also clear, with an overall decrease with size and local maxima for Au_5_^+^, Au_11_^+^ and particularly Au_15_^+^. Therefore, the interaction between Au*_n_*^+^ and Ar involves the transfer of electrons from Ar to Au, a more favorable process if the adsorption site has a higher positive charge. 

The bonding between Au*_n_*^+^ and Ar is further analyzed by calculating the Wiberg index (*W*) of the Au*_n_*^+^-Ar bond (structures in Figure 2), as presented in panel (b) of Figure 5. In line with the amount of electron charge from Ar to Au, the *W* index shows an overall decrease with *n*. For the smallest *n* = 3, 4 and 5 sizes the index is above 0.5, in line with the high Ar adsorption energies (above 0.3 eV). Interestingly, the Wiberg index has local maxima at Au_11_^+^ and Au_15_^+^ and a pronounced minimum at Au_16_^+^. The correlation between the partial positive charge of Ar and the Wiberg indices, together with the *W* values themselves being relatively large, in particular for *n* ≤ 5, reflects the formation of a chemical bond between Au*_n_*^+^ and Ar with a partial covalent character. A higher *W* index correlates perfectly with a smaller bond distance between Au and Ar in Au*_n_*Ar^+^, as shown in Figure 5b.

The density of states (DOS) of three selected clusters are presented in Figure 6: Au_3_Ar^+^, Au_15_Ar^+^ and Au_16_Ar^+^. In each case, the DOS was projected onto the states of the Au adsorption site and of Ar. For Au_3_Ar^+^, shown in the upper panel, there were three occupied states of major Ar character, which were highly hybridized with the states of the Au adsorption site (−16.4, −15.3 and −14.8 eV). Such overlap between the Ar and Au states clearly indicates the formation of a formal chemical bond between the atoms, in line with the high *W* index (0.62) for this cluster. Similar states were found in the PDOS of Au_15_Ar^+^ (middle panel), however in this case, the hybridization was smaller, as highlighted by the smaller *W* index (0.44). In contrast, the hybridization between Au and Ar is minor in Au_16_Ar^+^, in correspondence with the close to zero *W* index in this case, being only 0.24. Such a small *W* index is a consequence of the very limited electron charge transfer from Ar to Au in Au_16_Ar^+^, as seen from Figure 5a. 

The level of hybridization between the states of Au and Ar in the PDOS of Figure 6 are quantified by the Kullback–Leibler (*D_KL_*) divergence. The *D_KL_* divergence is a measure of how one distribution (*p*(*x*)) differs from another (reference) distribution (*q*(*x*)), and is defined as
(3)$$D_{KL}\left(p∥q\right)=\overset{\infty }{\underset{-\infty }{\int }}p\left(x\right)\ln\left(\frac{p\left(x\right)}{q\left(x\right)}\right)dx\;\;\;\;$$

The Kullback–Leibler divergence is zero when both distributions are identical, and increases the more they diverge from each other [39]. Using this analysis to quantify the hybridization between Au and Ar gives the values 3.18, 3.40 and 3.51, for Au_3_Ar^+^, Au_15_Ar^+^ and Au_16_Ar^+^, respectively. Therefore, the overlap between Au and Ar is larger in Au_3_Ar^+^, as expected, given its very high Ar adsorption energy, and the larger abundance in mass spectra. The overlap is smaller in Au_15_Ar^+^, but is still larger than in Au_16_Ar^+^. 

Hence, our analysis clearly shows the formation of a chemical bond between Au*_n_*^+^ and Ar. Such a bond, of significant covalent character, is strong for *n* ≤ 5, and presents an overall decrease with size. Au_11_Ar^+^ and particularly Au_15_Ar^+^ stand out in this trend. Au_15_^+^ adopts a geometry with a low-coordinated Au atom that possesses a high partial positive charge, favoring electron donation from Ar, thus enhancing the Au_15_^+^-Ar bonding. To the right of Figure 6, two molecular orbitals of Au_3_Ar^+^ (−16.4 and −15.3 eV orbitals), Au_15_Ar^+^ (−15.1 and −14.4 eV orbitals) and Au_16_Ar^+^ (−13.1 and −12.9 eV orbitals) are plotted. These orbitals clearly show the strong involvement of Ar in the wavefunctions of the Au_3_Ar^+^ and Au_15_Ar^+^ clusters, an atom that therefore cannot be considered inert when interacting with these Au*_n_*^+^ clusters. The involvement of Ar in the wavefunction of Au_16_Ar^+^ is much smaller. The binding analysis is complemented by calculations of the electron localization function (ELF), which corresponds to the probability of finding two electrons of the same spin in a determined region of space, and is therefore a useful quantity to differentiate between chemical and physical bonds. Calculations have shown ELF values above 0.1 in between atoms interacting covalently [40]. The analysis is presented in Figure 7 for Au_3_Ar^+^, Au_15_Ar^+^ and Au_16_Ar^+^, and reveals a relatively high ELF value between Au_3_^+^ and Ar, of 0.115. The ELF values decrease with size, being 0.086 and 0.056 for Au_15_Ar^+^ and Au_16_Ar^+^, respectively. 

## 5. Conclusions

In this work, the interaction of Ar with Au*_n_*^+^ clusters was investigated in the large *n* ≤ 20 size range, using a combination of mass spectrometry and density functional theory calculations. The abundances in mass spectra reveal a high propensity of the smaller clusters to form stable Au*_n_*Ar*_m_*^+^ complexes. In particular, large Ar fractions are seen in the *n* ≤ 7 range. With increasing cluster size, the fraction of adsorbed Ar decreases, and above *n* = 11, only complexes with a single Ar atom are formed. In this size range, Au_15_^+^ stands out with a very high fraction of Ar. The calculations reproduce outstandingly well the mass spectrometric observations. The formation of a chemical bond between Au and Ar is proven for the investigated complexes, and is shown to be a consequence of both the geometry adopted by Au*_n_*^+^ and their electronic structure. 

## Figures and Tables

**Figure 1 molecules-26-04082-f001:**
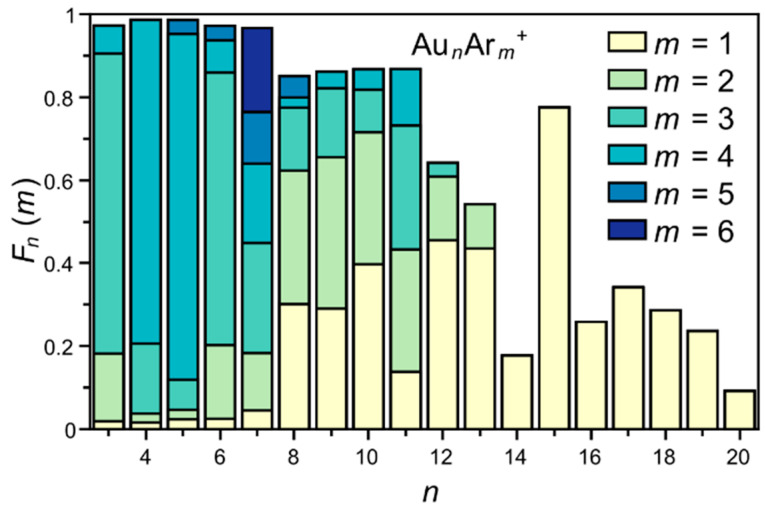
Fractional distribution of adsorbed Ar atoms in Au*_n_*Ar*_m_*^+^ (*n* = 3–20, *m* = 1–6) clusters, plotted as a stacked bar chart.

**Figure 2 molecules-26-04082-f002:**
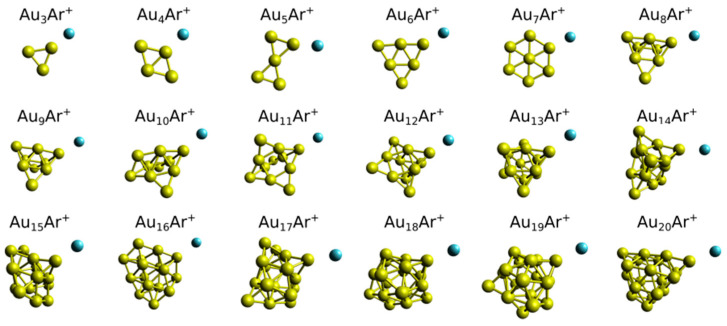
Optimized geometries of Au*_n_*Ar^+^ (*n* = 3–20) clusters, calculated at the PBE+D3/ECP(def2-TZVPP) level.

**Figure 3 molecules-26-04082-f003:**
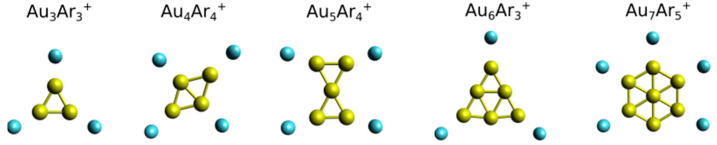
Optimized geometries of a selection of Au*_n_*Ar*_m_*^+^ (*n* = 3–7) clusters with multiple Ar atoms attached.

**Figure 4 molecules-26-04082-f004:**
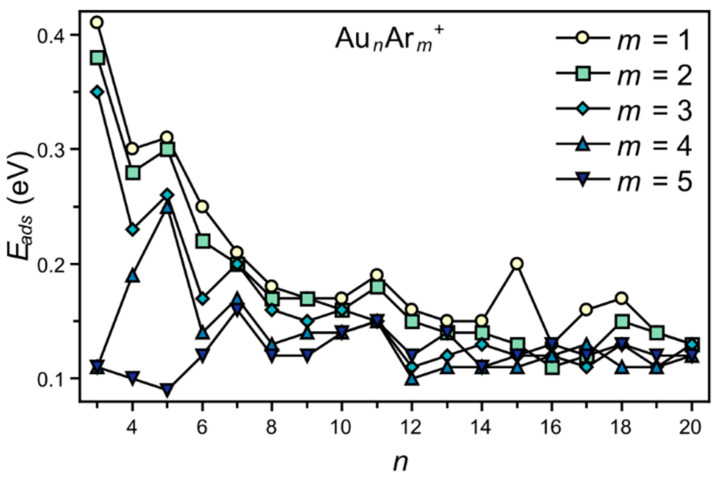
Adsorption energies of the *m*th adsorbed Ar atom in the Au*_n_*Ar*_m_*^+^ clusters, calculated using Equation (1).

**Figure 5 molecules-26-04082-f005:**
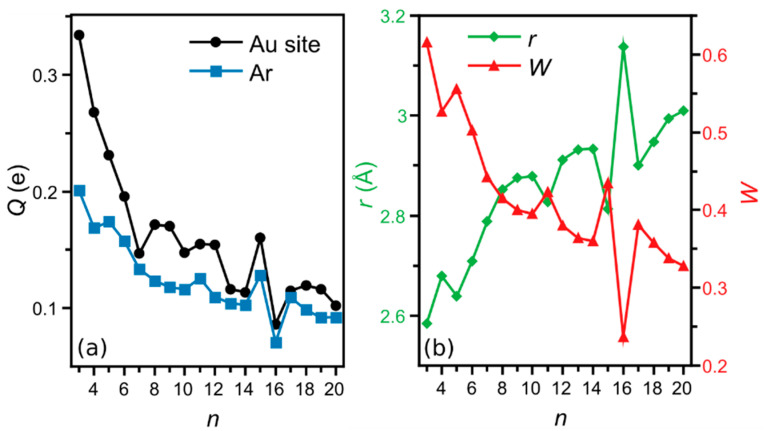
(**a**) Hirshfeld partial charge of the Au adsorption site of the bare Au*_n_*^+^ cluster (black circle), and Ar (blue square) in the Au*_n_*Ar^+^ complex. (**b**) Bond length (*r*—green diamonds) and Wiberg index (*W*—red triangles) of the Au*_n_*−Ar^+^ bond as a function of the cluster size *n*.

**Figure 6 molecules-26-04082-f006:**
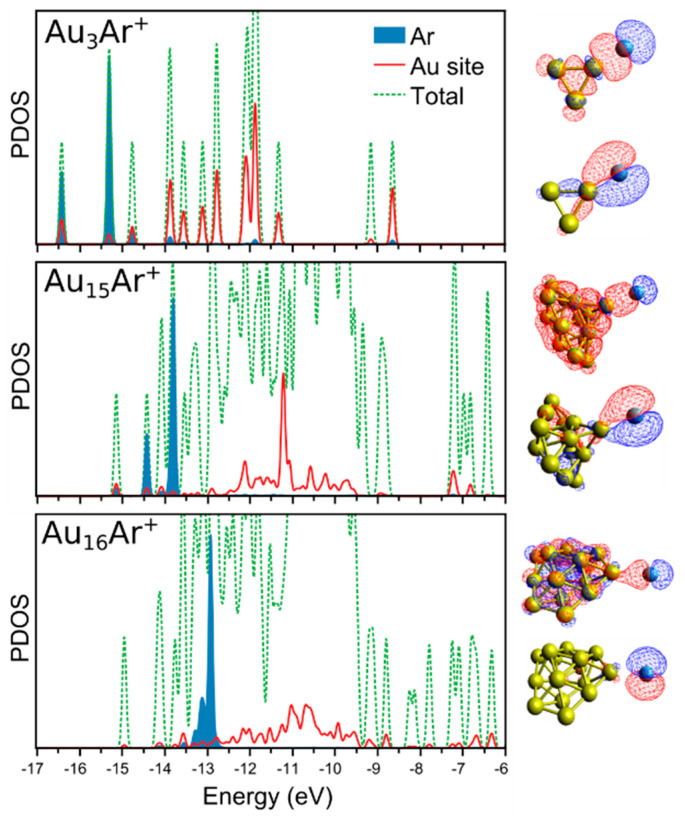
Density of states (DOS) of Au_3_Ar^+^, Au_15_Ar^+^ and Au_16_Ar^+^ clusters. The total DOS is presented by the dotted green lines, whereas the DOS projected (PDOS) onto the states of the Au adsorption site and the Ar atom are shown by the red lines and filled blue curves, respectively. At the right, plots of two selected Kohn−Sham molecular orbitals with major contribution from Ar are presented for each cluster. These orbitals are located at energies of −16.4 and −15.3 eV for Au_3_Ar^+^, −15.1 and −14.4 eV for Au_15_Ar^+^, and −13.1 and −12.9 eV for Au_16_Ar^+^, in the corresponding PDOS.

**Figure 7 molecules-26-04082-f007:**
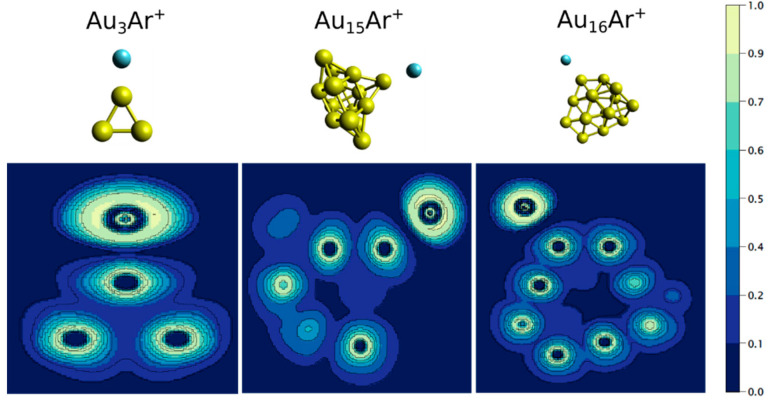
Electron localization function (ELF) of Au_3_Ar^+^, Au_15_Ar^+^ and Au_16_Ar^+^. At the top of each panel, the geometry of the cluster is presented.

## Data Availability

The coordinates of the optimized Au*_n_*Ar*_m_*^+^ (*n* = 3–20 and *m* ≤ 5) isomers are available through Github at https://github.com/pferrari13/Au_nAr_m-XYZ-coordinates.git.

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
