# Peer review of "Argon Adsorption on Cationic Gold Clusters Au*_n_*^+^ (*n* ≤ 20)"

_molecules, 2021, doi:10.3390/molecules26134082_

Round 1
Reviewer 1 Report
In this manuscript the authors presented the experimental results of the fractional distribution of adsorbed Ar on (Au)n+ clusters, and performed PBE-D3BJ/TZVPP calculations on the binding energies as well as Hirshfeld charge analysis and DOS analysis on some complexes. The results of the calculations supported the experimental observation, in particular the relative strong binding between Ar and Au15+. The quality of the calculation is reasonable, although more accurate wave function based methods and explicit inclusion of scalar relativistic effects are desired today for the smaller complexes. Overall, I can recommend the publication of the manuscript in Molecules, provided that the authors properly address the following:
-
At the end of section 4 the authors showed the picture a few molecule orbitals, presumably the canonical Kohn-Sham orbitals, as an evidence of bonding. However, these orbitals are often delocalized and not suitable as intuitive models of chemical bonding. Additional work, e.g., orbital localization, is usually needed in such interpretations. In addition, the orbitals from the relatively weakly bonded Au16Ar+ shall be provided as a comparison.
-
Some of the (Au)n(Ar)m+ complexes should have isomers due to different adsorption sites. Since mass spectroscopy does not provide any info on the structure, some results shall be provided to demonstrate that the energy differences between the isomers sufficiently small compared with the binding energy differences for different values of n/m.
Minor issues:
-
Are the computational method identical to that used in optimizing the Au clusters, and how were the results validated? Being able to describe the Au+-(Ar)m interaction does not guarantee that the method is suitable for describing Au clusters.
-
Are the binding energies reported ZPE corrected? If not what are the expected values?
-
It would be best that the xyz coordinates of the geometries of the complexes (that are not previously published) be provided in the supplementary materials, preferably in pure text files, so that other workers could investigate these systems and compare the results.
Author Response
1. At the end of section 4 the authors showed the picture of a few molecular orbitals, presumably the canonical Kohn-Sham orbitals, as an evidence of bonding. However, these orbitals are often delocalized and not suitable as intuitive models of chemical bonding. Additional work, e.g., orbital localization, is usually needed in such interpretation. In addition, the orbitals from the relatively weakly bonded Au16+ shall be provided as a comparison.
Reply: Figure 7 of the original manuscript indeed presented selected Kohn-Sham orbitals. In the revised manuscript, we have moved these orbitals to Figure 6, added those of Au16Ar+, and mention that it are Kohn-Sham orbitals. In addition, based on the comment of the referee, we performed a calculation of the electron localization function (ELF) for the Au3Ar+, Au15Ar+ and Au16Ar+ clusters. The ELF results are plotted in Figure 7 and the ELF values between Aun+ and Ar confirm the binding strength order n = 3 > 15 > 16. For Au3Ar+, a minimum value of 0.115 is found in the region between Au3+ and Ar, reflecting a relatively high electron density overlap. For purely physical interactions, ELF values are typically lower than 0.1 (K. Koumpouras, J. A. Larsson, J. Phys. Condens. Matter., 32, 315502, 2020).
2. Some of the (Au)n(Ar)m+ complexes should have isomers due to different adsorption sites. Since mass spectrometry does not provide any info on the structure, some results shall be provided to demonstrate that the energy differences between the isomers is sufficiently small compared with the binding energy differences for different values of n/m.
Reply: To elaborate on the referee’s remark, we have selected two small gold clusters with different Ar adsorption sites: Au4Ar+ and Au6Ar+. In the supporting information, the relative energies of three isomers with different Ar adsorption sites are presented. For Au4Ar+, the relative energies of the isomers are 0.05 and 0.20 eV, whereas for Au6Ar+ these are 0.08 and 0.12 eV. These energies are relatively high compared to the Ar binding energies of 0.30 and 0.25 eV for Au4Ar+ and Au6Ar+, respectively. Those relative energies imply that the isomer with the most stable Ar adsorption site will likely be dominant in the molecular beam, and if isomers are present, their population will be minor. Nevertheless, the possibility that isomers of other clusters sizes are present in the molecular beam, or that there is fluxionality between isomers, cannot be excluded, despite the very good agreement between our experimental results and the calculations. In the revised manuscript, we have included a statement about this possibility.
Minor issues:
- Are the computational methods identical to that used in optimizing the Au clusters, and how were the results validated? Being able to describe the Au+-(Ar)m interaction does not guarantee that the method is suitable for describing Au clusters.
Reply: The PBE/ECP(Def2-TZVPP) level of theory was indeed used in Ref. [20] to optimize the geometries of the bare Aun+ (n ≤ 9) clusters. In that reference, the level of theory was proven suitable for describing the structures of the gold clusters and a benchmark analysis was performed. That is actually the reason why we selected this computational method. We have made this clearer in the revised version of the manuscript.
- Are the binding energies reported ZPE corrected? If not what are the expected values?
Reply: In view of the extended size range that was investigated in the current work and the vibrational analysis being very demanding for the largest clusters, such as Au20Ar5+, we have not computed the vibrational modes and thus binding energies are not ZPE corrected. This is now explicitly mentioned in the methods section. Calculations on the small Au3Arm+ and Au4Arm+ clusters provide ZPE corrected adsorption energies that are 0.08 eV lower than without the ZPE correction. Importantly, the correction is rather independently on the number of Ar atoms per cluster and therefore, it does not change the conclusions drawn on the basis of Fig. 4.
- It would be best that the xyz coordinates of the geometries of the complexes (that are not previously published) be provided in the supplementary materials, preferably in pure text files, so that other workers could investigate these systems and compare the results.
Reply: Following the advice of the referee we decided to make the xyz coordinates available through a publically available data depository (https://github.com/pferrari13/Au_nAr_m-XYZ-coordinates.git).
Reviewer 2 Report
This work by Ferrari and Janssens investigated the adsorption behavior of Argon atoms onto the surface of cationic gold clusters with no more than 20 Au atoms. This manuscript presents basically the joint experimental and computational studies, however, several issues need to be addressed before being recommended for publication.
1) In the introduction part, the authors ignored several important review articles on the coinage metal clusters. For example, Coordination Chemistry Reviews, 2021, 429, 213643.
2) Is there any reason why the Hirshfeld partial charge was employed in this work while other types of atomic charge (e.g. NPA charge) were not used?
3) The authors employed density of states (DOS) in order to show that the interaction between Ar and gold cluster truly exists. Another option can be to use AIM theory to find the bond critical points, is there any reason why AIM was not used in this work?
4) For the supporting information, just providing the picture of complexes is not sufficient. One needs to put the Cartesian coordinates in the SI, so that readers may directly use them if they want to reuse the data in this work.
In general, this is an important work, but above issues need to be addressed.
Author Response
1. In the introduction part, the authors ignored several important review articles on the coinage metal clusters. For examples, Coordination Chemistry Reviews, 2021, 429, 213643.
Reply: The literature on coinage metal clusters is extensive. It has never been our intention to provide an exhaustive overview about coinage metal clusters or to comment in detail on the difference between gold, silver, and copper clusters. We followed the advice of the referee to guide interested readers to a few review papers on coinage metal clusters, including the one suggested by the referee.
2. Is there any reason why the Hirshfeld partial charge was employed in this work while other types of atomic charge (e.g. NPA charge) were not used?
Reply: There are many different schemes for describing atomic partial charges in a molecule, all having advantages and drawbacks. To analyze our results we opted for the Löwdin and the Hirshfeld methods. Qualitatively similar results were obtained with both methods (this is now mentioned in the manuscript). The Hirshfeld data was selected for discussion in the manuscript as the method is based on the electron density, instead of on molecular orbitals and therefore, is less basis-set dependent.
3. The authors employed density of states (DOS) in order to show that the interaction between Ar and gold cluster truly exists. Another option can be to use AIM theory to find the bond critical points, is there any reason why AIM was not used in this work?
Reply: A similar comment was made by Referee 1. We now computed the electron localization function (ELF) of the Au3Ar+, Au15Ar+ and Au16Ar+ clusters to analyze the metal cluster-argon bonding and added those results to the revised manuscript (see revised figure 7). The results of the ELFs support the earlier conclusions, the ELF values between Aun+ and Ar decrease from n = 3 over n =15 to n = 16. For Au3Ar+, a value of 0.115 is found in the region between Au3+ and Ar, showing a relatively high overlap between the electron densities of both particles. For purely physical interactions, ELF values lower than 0.1 have been reported (K. Koumpouras, J. A. Larsson, J. Phys. Condens. Matter., 32, 315502, 2020).
4; For the supporting information, just providing the picture of complexes is not sufficient. One needs to put the Cartesian coordinates in the SI, so that readers may directly use them if they want to reuse the data in this work.
Reply: Following the advice of the referee we decided to make the xyz coordinates available through a publically available data depository (https://github.com/pferrari13/Au_nAr_m-XYZ-coordinates.git).
Round 2
Reviewer 2 Report
The authors have addressed the issues properly. This work is recommended for publication.